# Preliminary Pilot Study of Combined Effects of Physical Activity and Achievement of LDL-Cholesterol Target on Coronary Plaque Volume Changes in Patients with Acute Coronary Syndrome

**DOI:** 10.3390/jcm9051578

**Published:** 2020-05-22

**Authors:** Miho Nishitani-Yokoyama, Katsumi Miyauchi, Kazunori Shimada, Takayuki Yokoyama, Shohei Ouchi, Tatsuro Aikawa, Mitsuhiro Kunimoto, Miki Yamada, Akio Honzawa, Shinya Okazaki, Hiroaki Tsujita, Shinji Koba, Hiroyuki Daida

**Affiliations:** 1Department of Cardiovascular Medicine, Graduate School of Medicine, Juntendo University, Tokyo 113-8421, Japan; mipocch@juntendo.ac.jp (M.N.-Y.); ktmmy@juntendo.ac.jp (K.M.); shimakaz@juntendo.ac.jp (K.S.); t-yoko@juntendo.ac.jp (T.Y.); uchi@juntendo.ac.jp (S.O.); taikawa@juntendo.ac.jp (T.A.); mkunimo@juntendo.ac.jp (M.K.); shinya@juntendo.ac.jp (S.O.); 2Cardiovascular Rehabilitation and Fitness, Juntendo University Hospital, Tokyo 113-8431, Japan; msioya@juntendo.ac.jp (M.Y.); sh4112035@juntendo.ac.jp (A.H.); 3Division of Cardiology, Department of Medicine, School of Medicine, Showa University, Tokyo 142-8666, Japan; h.tsujita@med.showa-u.ac.jp (H.T.); skoba@med.showa-u.ac.jp (S.K.); 4Faculty of Health Science, Juntendo University, Tokyo 113-0033, Japan

**Keywords:** acute coronary syndrome, cardiac rehabilitation, coronary plaque regression, physical activity, achievement of LDL-C target, integrated backscatter intravascular ultrasound

## Abstract

Background: We investigated the combined effects of physical activity (PA) and aggressive low-density lipoprotein cholesterol (LDL-C) reduction on the changes in coronary plaque volume (PV) in patients with acute coronary syndrome (ACS) using volumetric intravascular ultrasound (IVUS) analysis. Methods: We retrospectively analyzed data from two different prospective clinical trials that involved 101 ACS patients who underwent percutaneous coronary intervention (PCI) and assessed the non-culprit sites of PCI lesions using IVUS at baseline and at the follow-up. After PCI, all the patients participated in early phase II comprehensive cardiac rehabilitation. Patients were divided into four groups based on whether the average daily step count, measured using a pedometer, was 7000 steps of more and whether the follow-up LDL-C level was <70 mg/dL. At the time of follow-up, we examined the correlation of changes in the PV with LDL-C and PA. Results: The baseline characteristics of the four study groups were comparable. At the follow-up, plaque regression in both the achievement group (PA and LDL-C reduction) was higher than that in the other three groups. In addition, plaque reduction independently correlated with increased PA and reduction in LDL-C level. Conclusions: Combined therapy of intensive PA and achievement of LDL-C target retarded coronary PV in patients with ACS.

## 1. Introduction

The prognosis of acute coronary syndrome (ACS) patients is poor; however, the management of low-density lipoprotein cholesterol (LDL-C) levels improves the prognosis. The mechanism can be explained by plaque regression with lipid-lowering treatment [1]. Therefore, plaque regression following the management of blood lipid levels has clinical significance. However, managing LDL-C may not be sufficient to achieve maximum plaque reduction and the subsequent cardiac events [2]. One promising factor is that of physical exercise therapy [3].

Comprehensive Cardiac rehabilitation (CR) including exercise therapy after ACS is associated with lower cardiovascular mortality, re-hospitalization, coronary revascularization [3]. Moreover, the underlying mechanism is considered to be the anti-atherosclerosis effect of CR [3]. In fact, we recently used serial integrated backscatter intravascular ultrasonography (IVUS) to show that coronary plaque volume (PV) regression in ACS patients significantly and positively correlated to physical activity (PA) [4]. In addition, we demonstrated that intensive PA retarded coronary PV progression and ameliorated lipid component in patients with ACS participating in late phase II CR [5]. Therefore, intensive PA appears effective against residual risk for stabilization of plaque and reduction in clinical events. However, the combined effects of LDL-C reduction and PA on coronary PV have not been investigated, especially in ACS patients. Our study aimed to investigate the combined effects of PA and achievement of LDL-C target on the changes in coronary PV in ACS patients, as assessed using volumetric intravascular ultrasound (IVUS) analysis.

## 2. Methods

### 2.1. Study Subjects and Protocol

The current retrospective study evaluated the efficacy of PA and LDL-C reduction in the database for two different clinical trials. One of these trials aimed to determine the effects of phase II CR on coronary PV in patients after ACS [4]. The other trial examined the effects of CR involving intensive PA on coronary PV and components in patients with ACS [5]. The details of those two trials have been previously described [4,5]. In brief, all the patients were admitted with ACS and underwent emergency coronary stenting in a culprit lesion. At the beginning of the study and at the time of follow-up (at 6 to 8 months after the percutaneous coronary intervention (PCI)), the serial volumetric IVUS examination was performed to assess the changes in coronary plaque in the non-culprit lesion. The target segment was determined at a non-PCI site (5 mm proximal or distal to the PCI site) of culprit vessel with a reproducible index, usually a branch site, on the PCI vessel [4,5]. After the procedure, all the patients participated in early-phase II CR. We recommended that all the patients participate in late-phase II CR after discharge and to perform walking exercise as per the exercise prescription. Follow-up visits were scheduled every 4 weeks after PCI. We checked their exercise regimen and daily PA at every visit. Baseline measurements included IVUS data and biochemical data on admission as well as cardiopulmonary exercise test (CPX) on discharge. Follow-up IVUS data, biochemical data, and CPX were repeated at the time of follow-up. At the follow-up IVUS, the LDL-C target for post-ACS management was defined as LDL-C < 70 mg/dL [6].

Based on the serial IVUS findings of the target segment relative to the baseline, plaque regression was defined as a reduction in the absolute change in the percent atheroma volume (PAV), while plaque progression was defined as a rise in the absolute change in the PAV.

This study was conducted as per the Declaration of Helsinki. All the patients provided written informed consent, and the Ethics Committee of each institution approved the study (the approval numbers from the Ethics Committees of Juntendo University Hospital and Showa University Hospital were ID 24-018, 17-171, and ID 1036, respectively). The study was registered in the UMIN Clinical Trials Registry System under trial ID UMIN000006038 and 000010031.

### 2.2. Early Phase II CR Protocol

The CR program, including warm-up stretching, aerobic exercise, and cool-down, was scheduled once a day in the hospital, as described previously [4,5]. Aerobic exercise comprised the use of a cycle ergometer and treadmill as well as walking on an in-cardiac rehabilitation room track. The total aerobic exercise time was about 20 min. The exercise intensity was prescribed individually at the anaerobic threshold (AT) level, as measured using an ergometer test using expiratory gas analysis and a rating of 11–13 on the standard Borg’s perceived exertion scale [7]. All the subjects were instructed to follow the phase II diet of the American Heart Association at the initiation of the CR program [8]. At baseline, an educational program was provided for each subject by physicians, nurses, and dietitians regarding coronary artery disease (CAD) and its risk factors.

### 2.3. Measurements

We assessed the body composition and exercise tolerance at baseline and at the time of follow-up. Anthropometric parameters were assessed using the body mass index (BMI) and body composition. The percentages of body fat and lean body weight were measured with a TANITA MC-780A^®^ Body Composition Analyzer (TANITA Co. Ltd., Tokyo, Japan) that performed measurements in the state where it stood on the electrode based on bioelectrical impedance analysis. Derivation of body volume, together with measurement of body mass, permits the calculation of body density and subsequent estimation of percent fat and fat-free mass. In order to measure the peak oxygen consumption (peak VO2) and oxygen uptake at the AT, patients were subjected to ergometer testing (Strength Ergo8, NIHON KOHDEN Co. Ltd., Tokyo, Japan) using an expiratory gas analysis machine (AE-310s, Minato Medical Science Co. Ltd., Osaka, Japan). After a period of resting, warm-up was performed for few minutes at 0 W, followed by ramp loading (10 W/min) till the subject was exhausted, got progressive angina, developed ST-segment depression (≥2 mm), or sustained tachyarrhythmia. The point of AT was determined using the “V-slope” method. Daily physical activity was evaluated using a pedometer (Lifecoder^®^, Suzuken Co, Ltd., Nagoya, Japan) that could record the mean step count and calculate the mean energy of PA for up to 60 days.

### 2.4. Statistical Analyses

The results are expressed as mean value ± standard error values and were compared using one-way analysis of variance. Comparisons of the continuous variables of the 4 groups were performed with analysis of variance (ANOVA). The absolute change between the baseline and follow-up data was assessed using paired *t*-test or Wilcoxon signed rank test, according to their distributions. The correlation coefficients were determined using linear regression analysis. Stepwise multiple regression analysis was used determine the independent predictors of the changes in PV. Statistical significance of the correlation coefficients was determined using the method of Fisher and Yates. A *p*-value < 0.05 was considered significant. All the data were analyzed using JMP software (Version 14 for Windows, SAS Institute Inc. Cary, NC, USA).

## 3. Results

### 3.1. Study Population and Clinical Characteristics

One of these trials enrolled 50 ACS patients who were admitted to the Juntendo University Hospital and Showa University Hospital from December 2009 to August 2012 [4]. The other trial enrolled 51 ACS patients who were admitted to the Juntendo University Hospital from February 2013 to January 2016 [5]. We analyzed the data of 101 patients. Overall, the mean patient was 60.7 ± 10.2 (mean value ± standard deviation) years; 95% were men. The prevalence of coronary risk factors was as follows: hypertension, 62%; diabetes mellitus, 43%; and dyslipidemia, 91%. Seventy-four percent of the patients had ST-elevation myocardial Infarction (STEMI).

Of the 101 patients, 2 patients could not continue PA due to orthopedic disease, 2 patients dropped out of the follow-up, and 35 patients were excluded due to the poor image quality of IVUS. In the present study, the patients were divided into four groups according to whether the mean step counts measured using the pedometer was 7000 steps (median value) or more and the LDL-C level at follow-up was <70 mg/dL. Finally, 62 patients were enrolled and divided into the 4 groups as follows. Group 1: on-treatment, LDL-C ≥ 70 mg/dL, daily mean steps < 7000, *n* = 26; group 2: on-treatment, LDL-C < 70 mg/dL, daily mean steps < 7000, *n* = 9; group 3: on-treatment, LDL-C ≥ 70 mg/dL, daily mean steps ≥ 7000, *n* = 17; and group 4: on-treatment, LDL-C < 70 mg/dL, daily mean steps ≥ 7000, *n* = 10 (Figure 1). The clinical data, including IVUS measurements at baseline and follow-up, of all the patients were recorded.

The baseline clinical characteristics of the subjects and medication are presented in Table 1. There was a significant difference in the age of the subjects in the four groups. No significant differences were present in the distribution of sex, coronary risk factors, ACS classification, number of diseased vessels, the levels of the ejection fraction, and medication.

### 3.2. Physical Activity in Each Patient during Outpatient Clinic Visit

Figure 2 shows PA during 60 days after discharge in each patient. Of the 101 patients, 2 patients could not continue physical activity because of orthopedic disease and 2 patients dropped out during follow-up. Therefore, we present the data of 97 patients who could be evaluated in terms of PA records using pedometers. In the overall population, the median PA was 6933 steps/day (interquartile range (IQR) 4500–8318 steps/day). Therefore, in further analyses, we divided the study subjects into the following two groups as per the median PA value of 7000 steps; active group: *n* = 27, daily PA ≥ 7000 steps and inactive group: *n* = 35, daily PA < 7000 steps). At the follow-up, the inactive group had significant improvements in PA (both *p* < 0.01). The active group maintained their PA during this period. Among the four groups, the levels of PA were significantly different at follow-up (*p* < 0.05) (Figure 3).

### 3.3. Serum Lipid Profile, Glucose Parameters, Anthropometric Parameters, and Exercise Tolerance of the Subjects of the 4 Groups at Baseline and Follow-Up

Serum lipid profiles, glucose parameters, anthropometric parameters, and exercise tolerance at baseline at follow-up are shown in Table 2. The mean baseline and follow-up LDL-C values were 128 ± 45 (mean value ± standard deviation) mg/dL and 76 ± 20 (mean value ± standard deviation) mg/dL, respectively. In the overall population, the absolute change in the LDL-C was −50 ± 44 (mean value ± standard deviation) mg/dL, and the percent change in LDL-C was −34.5 ± 24.2 (mean value ± standard deviation) %. At the follow-up, the LDL-C levels of 31% of the patients fell below 70 mg/dL. The LDL-C levels and BMI were significantly different among the four groups at baseline. The levels of high-density lipoprotein cholesterol (HDL-C), triglycerides, glycosylated hemoglobin (HbA1c), and exercise tolerance at baseline were not significantly different among the four groups. At the follow-up, the LDL-C levels were significantly lower than those at baseline in each of the four groups (*p* < 0.05) (Figure 3). The levels of HDL-C, triglycerides, HbA1c, and BMI were not significantly different in the absolute changes at follow-up.

The fat weight and lean body weight of the four groups were not significantly different at baseline and at the follow-up. At baseline, the peak VO_2_ levels of the four groups were similar. At the follow-up, each group showed significant improvements in exercise tolerance (respectively, *p* < 0.05).

### 3.4. Volumetric Analysis of the IVUS Parameters

The results of the volumetric analyses of the IVUS parameters are presented in Table 3. There were no significant differences in the vessel volume, lumen volume, or PV at baseline among the four groups. The PV was reduced at the follow-up examination in the active groups (both *p* < 0.05). However, The PV was not reduced at the follow-up examination in the inactive groups (both *p* < 0.05). The percent change in the PV significantly differed among the four groups (group 1; −2.1 ± 2.1 vs. group 2; 0.3 ± 3.6 vs. group 3; −9.6 ± 2.6 vs. group 4; −12.3 ± 3.4%, *p* = 0.01) (Figure 4).

### 3.5. Correlations between Plaque Changes and PA

The PA (*r* = −0.45, *p* < 0.01) and the changes in the LDL-C level (*r* = −0.30, *p* = 0.01) demonstrated significant negative correlations with the percent changes in PV. Therefore, we performed multivariable linear regression analyses as follows. In a multivariate analysis that included age, BMI, presence/absence of diabetes, fasting glucose level, change in LDL-C level, change in HDL-C level, PA, change in oxygen consumption, and baseline PV, PA (β = −0.34, *p* = 0.02), and the change in LDL-C level (β = −0.30, *p* = 0.03) were significantly associated with the change in percent PV (Table 4).

## 4. Discussion

The major findings of the present study were as follows: (1) The combination of higher PA and reduced LDL-C prevented atherosclerosis development; (2) the percent change in PV was significantly and independently correlated with PA and reduction in the LDL-C level. To our knowledge, this is the first report that has investigated the combined effects of PA and achievement of LDL-C target on coronary PV in patients with ACS, as assessed using IVUS.

ACS patients have multiple lesions with instability plaques. ACS is known to develop from mild atherosclerotic lesions with vulnerability plaque [9]. Furthermore, LDL-C is a key modifiable risk factor in plaque progression [10,11]. In patients with coronary artery disease, the prevalence of cardiovascular events is lower in those who achieve very low LDL-C levels than in those who achieve moderately low LDL-C levels [12]. In 2017, the Japan Atherosclerosis Society issued guidelines which recommended a target of LDL-C level < 70 mg//dL for secondary prevention in ACS patients [6]. We have reported that early aggressive lipid-lowering therapy with atorvastatin for 6 months significantly reduced the plaque volume in ACS patients [13]. Furthermore, the JAPAN-ACS study that was based on a randomized, large-scale, multicenter study demonstrated that intensive statin therapy in ACS patients showed significant regression of coronary PV [14]. Therefore, it is crucial to reduce the LDL-C level aggressively using statins [15].

There are many important coronary risk factors in addition to LDL-C level, such as smoking, diabetes, dyslipidemia, hypertension, and physical inactivity. These factors are the residual treatment targets in ACS patients. The CR program is a long-term, comprehensive secondary prevention program that involves medical assessment, exercise prescription, coronary risk factor control, education, and counseling for patients with cardiovascular disease. [16,17]. The guideline of the Japanese Circulation Society recommends moderate-to-vigorous-intensity aerobic exercise training 3 times a week and for 30 min per session in ACS patients [18]. We have already reported that intensive PA retarded and stabilized coronary PV in two different clinical trials [4,5]. A previous study demonstrated that a combination of statin therapy and lifestyle modification—including PA—inhibited angiographic minimal lumen stenosis [19]. However, to our knowledge, this is the first study on the combined effects of PA and LDL-C reduction on coronary PV changes using IVUS.

There are several mechanisms through which intensive PA and the achievement of target LDL-C level ameliorate coronary PV in ACS patients. Inflammatory markers—such as C-reactive protein level, interleukin-6 level, and proinflammatory cytokine—are major factors that influence the development of ACS [20,21]. Kurose et al. reported a decrease in high sensitive C-reactive protein as an independent predictor of coronary plaque regression in ASC patients who participated in CR [22]. We reported that CR for 6 months ameliorated not only metabolic parameters, but also the exercise capacity, muscle strength, and inflammatory state in metabolic syndrome patients after CABG [23]. We demonstrated that the IL-6 level was low in the active group than in the inactive group and was significantly and inversely correlated with exercise tolerance and PA at follow-up [5]. These results suggest that regular PA is associated with a reduction in the level of inflammatory markers. Exercise training improves endothelium-dependent vasodilatation both in epicardial coronary vessels and in resistance vessels through the enhancement of endothelial nitric oxide synthase and the reduction of reactive oxygen species in CAD patients [24]. These pathways were partially explained by the present results. Hambrecht et al. reported that the regression of coronary atherosclerotic lesions was observed in CAD patients who burnt an average of 2200 kcal/week with regular physical exercise [25]. In this study, we prescribed an intensity of exercise of almost 3.3 metabolic equivalents (METs) to patients evaluated using the CPX at discharge, and the calculated mean energy expenditure of PA in the active group was 2140 ± 202 kcal/week.

More than 30 years after the launch of statins, the strategy of “the lower, the better” for LDL-C is well established. Medical costs associated with cardiovascular disease are increasing considerably across the globe. Therefore, an efficacious, cost-effective therapy, such as CR, that allows the effective use of medical resources, is vital [26]. Therefore, the CR program, including life modification in addition to LDL-C reduction could be a suitable treatment strategy for “beyond LDL” for the residual cardiovascular risk in ACS patients.

There are certain limitations to the present study. First, this study has a relatively small sample size, because it is a preliminary and pilot study. Studies with a larger sample size are needed to confirm these findings. Second, a pedometer is unable to provide information regarding non-walking related activities. Therefore, the evaluation of the daily activity might have been underestimated.

## 5. Conclusions

The combined effects of intensive PA and the achievement of the LDL-C target ameliorate coronary PV in ACS patients. The CR program, including intensive PA and reduction in the LDL-C level, could be a suitable treatment strategy “beyond LDL” for the residual cardiovascular risk in ACS patients.

## Figures and Tables

**Figure 1 jcm-09-01578-f001:**
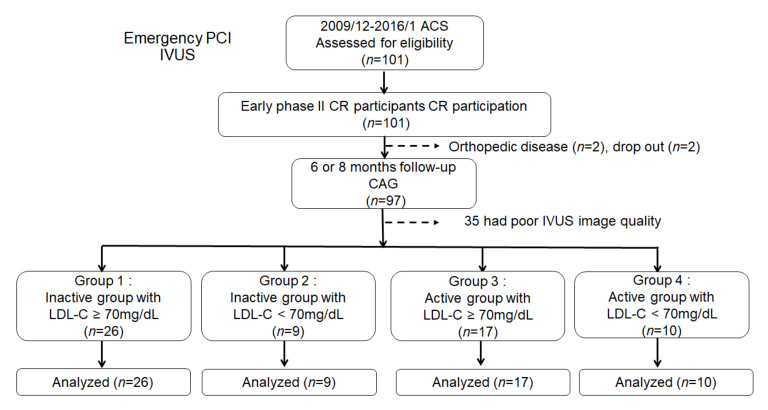
Study flow chart. Inactive group: Physical activity < 7000 steps/day, Active group: Physical activity ≥ 7000 steps/day. PCI, percutaneous coronary intervention; IVUS, intravascular ultrasound; ACS, acute coronary syndrome; CR, cardiac rehabilitation; LDL-C, low-density lipoprotein cholesterol; CAG, coronary angiography.

**Figure 2 jcm-09-01578-f002:**
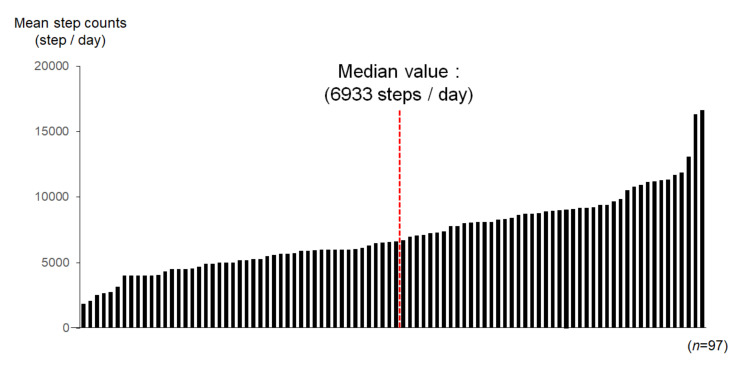
Physical activity in each patient assessed during the outpatient clinic visit.

**Figure 3 jcm-09-01578-f003:**
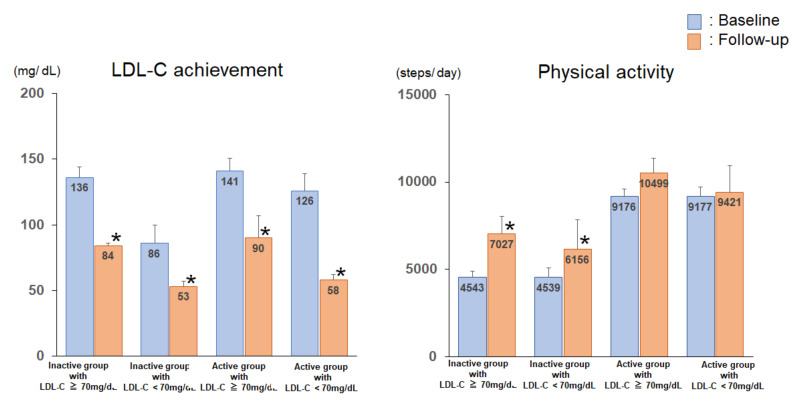
Comparison of the LDL-C achievement and physical activity among the four groups at baseline and at the follow-up. * *p* < 0.05 compared with the value at baseline.

**Figure 4 jcm-09-01578-f004:**
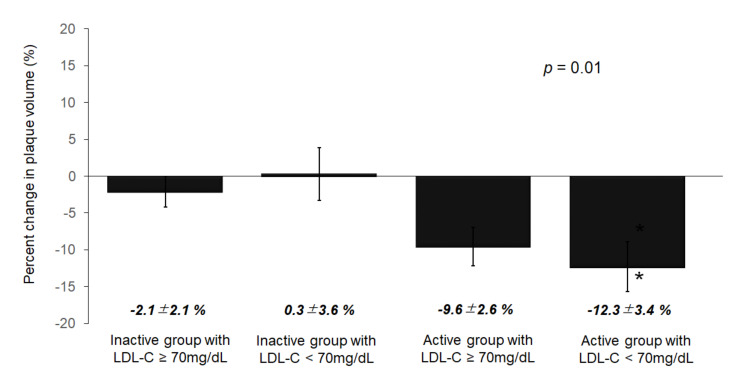
Percent change in the plaque volume among the four groups. The percent change in PV significantly differed among the 4 groups (group 1; −2.1 ± 2.1 vs. group 2; 0.3 ± 3.6 vs. group 3; −9.6 ± 2.6 vs. group 4; −12.3 ± 3.4%, *p* = 0.01). In particular, the percent change in PV in group 4 was significantly reduced (* *p* = 0.03 compared among 4 groups using Dunnett test). PV, plaque volume.

**Table 1 jcm-09-01578-t001:** Baseline patient characteristics.

	Inactive Group with LDL-C ≥ 70 mg/dL (*n* = 26)	Inactive Group with LDL-C < 70 mg/dL (*n* = 9)	Active Group with LDL-C ≥ 70 mg/dL (*n* = 17)	Active Group with LDL-C < 70 mg/dL (*n* = 10)	*p*-Value
Age, y	62.5 ± 1.9	67.0 ± 3.2	56.6 ± 2.3	55.9 ± 3.0	0.02
Male (%)	23 (88)	9 (100)	17 (100)	9 (90)	0.37
Hypertention (%)	16 (62)	6 (67)	10 (59)	7 (70)	0.93
Diabetes mellitus (%)	9 (35)	3 (33)	10 (59)	5 (50)	0.39
Dyslipidemia (%)	22 (85)	7 (78)	17 (100)	10 (100)	0.13
Current smoking (%)	12 (46)	4 (44)	8 (47)	7 (70)	0.79
Family history (%)	10 (38)	1 (11)	6 (35)	5 (50)	0.33
Classification of ACS (%)					
ST elevated MI (%)	20 (77)	6 (67)	8 (47)	5 (50)	0.28
Non-ST elevated MI (%)	6 (23)	2 (22)	7 (41)	3 (30)	
Unstable angina (%)	0 (0)	1 (11)	2 (12)	2 (20)	
Diseased vessels (%)					
Single	16 (62)	7 (78)	13 (76)	5 (50)	0.29
Double	10 (38)	2 (22)	4 (24)	4 (40)	
Triple	0 (0)	0 (0)	0 (0)	1 (10)	
Culprit vessel (%)					
RCA	12 (46)	3 (33)	5 (29)	2 (20)	0.56
LAD	10 (39)	6 (67)	9 (53)	7 (70)	
LCX	4 (15)	0 (0)	3 (18)	1 (10)	
Maximum CK, IU/L	2008 ± 374	1799 ± 637	845 ± 463	1960 ± 604	0.24
EF, %	55.6 ± 1.9	56.6 ± 3.7	55.0 ± 2.3	55.1 ± 3.1	0.97
Medications, *n* (%)					
DAPT (%)	26 (100)	9 (100)	17 (100)	10 (100)	0.96
ACE-I (%)	16 (62)	3 (33)	13 (76)	6 (60)	0.05
ARB (%)	7 (27)	5 (56)	4 (24)	3 (30)	0.41
β-blockers (%)	19 (73)	9 (100)	15 (88)	8 (80)	0.08
CCB (%)	4 (15)	1 (11)	2 (12)	2 (20)	0.70
Statin (%)	25 (96)	8 (89)	16 (94)	10 (100)	0.71
Ezetimibe (%)	2 (10)	0 (0)	2 (12)	3 (33)	0.08
α-GI (%)	4 (15)	0 (0)	2 (12)	0 (0)	0.70
SU (%)	0 (0)	0 (0)	0 (0)	1 (10)	0.12
Insulin (%)	0 (0)	1 (11)	1 (6)	1 (10)	0.45

Data are presented as the mean value ± standard error value. The figures in brackets are percentages. LDL-C, low-density lipoprotein cholesterol; CAD, coronary artery disease; ACS, acute coronary syndrome; MI, myocardial infarction; RCA, right coronary artery; LAD, left anterior descending artery; LCX, left circumflex artery; CK, creatine kinase; EF, ejection fraction; DAPT, dual antiplatelet therapy; ACE-I, angiotensin converting enzyme inhibitors; ARB, angiotensin II receptor blocker; CCB, calcium-channel blockers; α-GI, α-glucosidase inhibitors; SU, sulfonylurea.

**Table 2 jcm-09-01578-t002:** Comparison of anthropometric parameters, lipid profile and exercise tolerance among the four groups at baseline and follow-up.

	Inactive Group with LDL-C ≥ 70 mg/dL (*n* = 26)		Inactive Group with LDL-C < 70 mg/dL (*n* = 9)		Active Group with LDL-C ≥ 70 mg/dL (*n* = 17)		Active Group with LDL-C < 70 mg/dL (*n* = 10)	
	Baseline	Follow-Up	Baseline	Follow-Up	Baseline	Follow-Up	Baseline	Follow-Up
Anthropometric parameters								
Body mass index (kg/m^2^)	24.6 ± 0.6	23.7 ± 1.0	22.1± 1.1	24.3 ± 1.8	25.2 ± 0.8	24.9 ± 1.1	26.9 ± 1.0	21.4 ± 2.1
Fat weight (%)	27.8 ± 2.2	25.7 ± 1.9	18.9 ± 4.1	25.7 ± 3.6	23.8 ± 2.2	22.2 ± 2.0	32.0 ± 5.0	25.9 ± 4.4
Lean body weight (kg)	49.0 ± 2.8	51.2 ± 2.7	55.0 ± 5.2	52.3 ± 4.9	55.5 ± 2.8	53.5 ± 2.8	48.0 ± 6.3	50.6 ± 6.1
Exercise tolerance								
Peak VO_2_, mL kg^−1^ min^−1^	15.4 ± 0.5	18.9 ± 1.1 *	15.6 ± 0.9	19.1 ± 1.9 *	16.4 ± 0.6	21.8 ± 1.1 *	15.8 ± 0.8	20.0 ± 2.2 *
Lipid profile and glucose metabolism								
LDL-C, mg/dl	136 ± 8	84 ± 2 *	86 ± 14	53 ± 4 *	141 ± 10	90 ± 3 *	126 ± 13	58 ± 4 *
HDL-C, mg/dl	41 ± 2	40 ± 2	46 ± 4	45 ± 4	44 ± 3	45 ± 3	42 ± 3	42 ± 3
TG, mg/dl	154 ± 16	158 ± 11	98 ± 26	111 ± 18	153 ± 19	127 ± 13	179 ± 25	127 ± 17
FBS, mg/dl	115 ± 7	98 ± 2	108 ± 11	101 ± 5	108 ± 9	98 ± 3	143 ± 11	100 ± 4
Hemoglobin A1c, %	5.8 ± 0.1	5.6 ± 0.1	5.6 ± 0.2	5.8 ± 0.2	5.7 ± 0.2	5.6 ± 0.1	6.4 ± 0.2	6.0 ± 0.1

Data are presented as the mean value ± standard error value. * *p* < 0.05 compared with baseline. LDL-C, low-density lipoprotein cholesterol; HDL-C, high-density lipoprotein cholesterol; TG, triglyceride; FBS, fasting blood sugar.

**Table 3 jcm-09-01578-t003:** Comparison of IVUS data among the four groups.

	Inactive Group with LDL-C ≥ 70 mg/dL (*n* = 26)	Inactive Group with LDL-C < 70 mg/dL (*n* = 9)	Active Group with LDL-C ≥ 70 mg/dL (*n* = 17)	Active Group with LDL-C < 70 mg/dL (*n* = 10)	*p*-Value
IVUS profile at baseline					
Vessel volume, mm^3^	212.1 ± 26.4	163.6 ± 44.9	197.1 ± 32.7	173.4 ± 42.6	0.76
Lumen volume, mm^3^	113.2 ± 14.6	91.3 ± 24.9	102.3 ± 18.1	88.6 ± 23.6	0.78
Plaque volume, mm^3^	98.8 ± 12.8	72.2 ± 21.9	94.8 ± 15.9	84.7 ± 20.7	0.74
IVUS profile at follow-up					
Vessel volume, mm^3^	208.1 ± 25.8	161.6 ± 43.9	186.8 ± 31.9 *	150.0 ± 41.6 *	0.69
Lumen volume, mm^3^	111.4 ± 14.3	92.4 ± 24.3	101.6 ± 17.7	84.6 ± 23.1	0.76
Plaque volume, mm^3^	96.6 ± 12.3	69.1 ± 20.9	85.2 ± 15.2 *	74.4 ± 19.8 *	0.63
Percent change of VV, %	−2.4 ± 2.0	0.0 ± 3.4	−6.4 ± 2.5	−8.5 ± 3.3	0.20
Percent change of LV, %	−1.5 ± 3.3	1.9 ± 5.6	−4.0 ± 4.1	−3.6 ± 5.3	0.84
Percent change of PV, %	−2.1 ± 2.1	0.3 ± 3.6	−9.6 ± 2.6	−12.3 ± 3.4	0.01

Data are presented as the mean value ± standard error value. * *p* < 0.05 compared to at baseline. IVUS, intravascular ultrasound; VV, Vessel volume; LV, Lumen volume; PV, plaque volume.

**Table 4 jcm-09-01578-t004:** Multiple linear regression analysis of factors on the percent change of plaque volume.

	Univariate Analysis		Multivariate Analysis	
	*β*	*p*-Value	*β*	*p*-Value
Age	0.3	0.01		
Body mass index	−0.09	0.46		
Diabetes (absent-present)	−0.07	0.54		
Fasting glucose	0.04	0.76		
Delta LDL-C	0.3	0.01	0.3	0.03
Delta HDL-C	−0.21	0.1		
Physical activity	−0.45	0.0003	−0.34	0.02
Delta peak VO_2_	−0.09	0.44		
Plaque volume at baseline	0.09	0.50		

β: standardized partial regression coefficient. LDL-C, low-density lipoprotein cholesterol; HDL-C, high-density lipoprotein cholesterol; VO_2_: oxygen consumption.

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
