# Peer review of "Preliminary Pilot Study of Combined Effects of Physical Activity and Achievement of LDL-Cholesterol Target on Coronary Plaque Volume Changes in Patients with Acute Coronary Syndrome"

_jcm, 2020, doi:10.3390/jcm9051578_

Round 1

Reviewer 1 Report

Review 2:

  • Title should include the words “Preliminary Pilot Study”

Preliminary Pilot Study of Combined Effects of Physical Activity and 2 Achievement of LDL-cholesterol Target on Coronary 3 Plaque Volume Changes in Patients with Acute 4 Coronary Syndrome

  • The authors have not dealt with all my points. For example On the topic of plaque regression with LDL C reduction, the same group of authors have already published a summary of the research to date on this finding. However I cannot find this paper in their list of references for: J Atheroscler Thromb. 2019 Jul 1; 26(7): 592–600. doi: 10.5551/jat.48603.   PMCID: PMC6629749. The Goal of Achieving Atherosclerotic Plaque Regression with Lipid-Lowering Therapy: Insights from IVUS Trials.  Hiroyuki Daida, Tomotaka Dohi, Yoshifumi Fukushima, Hirotoshi Ohmura, and Katsumi Miyauchi
  • Table 2 seems unchanged and is very complex. The authors have not indicated that they revised it in line with my recommendations.

Kindly ask the authors to go through all of my recommendations and indicate in red where they have altered the document or sought advice.

If this is done then I am happy to review the paper again with all points dealt with and indicated clearly in the text where changes have been made.

Author Response

Response to Reviewer 1 Comments
Thank you very much for your valuable comments. We have attempted to address the questions and comments as follows:
Point 1: Title should include the words “Preliminary Pilot Study” Preliminary Pilot Study of Combined Effects of Physical Activity and Achievement of LDL-cholesterol Target on Coronary Plaque Volume Changes in Patients with Acute Coronary Syndrome
Response 1:
Thank you again very much for your terrific suggestion. We have revised the title, as follows:
Line 2-5:
From: Combined Effects of Physical Activity and Achievement of LDL-cholesterol Target on Coronary Plaque Volume Changes in Patients with Acute Coronary Syndrome
To: Preliminary Pilot Study of Combined Effects of Physical Activity and Achievement of LDL-cholesterol Target on Coronary Plaque Volume Changes in Patients with Acute Coronary Syndrome
Point 2: The authors have not dealt with all my points. For example On the topic of plaque regression with LDL C reduction, the same group of authors have already published a summary of the research to date on this finding. However I cannot find this paper in their list of references for: J Atheroscler Thromb. 2019 Jul 1; 26(7): 592–600. doi: 10.5551/jat.48603. PMCID: PMC6629749. The Goal of Achieving Atherosclerotic Plaque Regression with Lipid-Lowering Therapy: Insights from IVUS Trials. Hiroyuki Daida, Tomotaka Dohi, Yoshifumi Fukushima, Hirotoshi Ohmura, and Katsumi Miyauchi
Response 2:
Thank you very much for your suggestion. we have added the references in Discussion as follows:
Line 249:
From: Therefore, it is crucial to reduce the LDL-C level aggressively using statins.
To: Therefore, it is crucial to reduce the LDL-C level aggressively using statins [15].
[15] Daida H. et al. J Atheroscler Thromb. 2019 Jul 1; 26(7): 592–600.doi:10.5551/jat.48603.
Point 3: Table 2 seems unchanged and is very complex. The authors have not indicated that they revised it in line with my recommendations.
Response 3:
Thank you very much for pointing this out. As you point out, we have revised the table 2.

Reviewer 2 Report

Thank you for your edits. I have no major concerns.

Author Response

Thank you for your reviewing.

This manuscript is a resubmission of an earlier submission. The following is a list of the peer review reports and author responses from that submission.

Round 1

Reviewer 1 Report

The authors have done a considerable amount of work in the area and must be congratulated for this.

However, before submitting your paper for international peer review, please have it proof read to ensure that the standard of English used is appropriate. 

It is an interesting preliminary study. However due to the small numbers of patients and the resultantly large confidence intervals, I believe that it should be clearly labelled as a preliminary study

Also I have recommended that the figures are reviewed by a statistician as I have difficulty in seeing how, with such small numbers and large confidence intervals, the findings can be statistically significant. There clinical significance is another matter that is not addressed in the paper.

I attach my critique along with the paper and some annotations.

Reviewer 2 Report

Overall very good study with a nice comparison of lipids and activity. An interesting note that activity seemed to make a more important difference to regression than LDL when you look at figure 4 as both PA groups have regression.

Minor edits below

  1. In abstract the lines 25-27 (starting with Patients were…. 4 groups” should be in the methods section. It would be ideal to put in the PAV change in the combination group vs the other groups and the difference mentioned here.
  2. The sentence might read better as:

However, managing LDL-C may not be sufficient to achieve maximum plaque reduction and the subsequent cardiac events.

  1. Line 43. Remove the word And
  2. Line 64; Perform (not performe)
  3. Line 123: Should read
    In the 101 patients, 2 patients could not continue PA due to orthopedic disease, 2 patients dropped out of follow-up and 35 patients were excluded due to the poor image quality of IVUS.